# COVID-19 Pandemic: Escape of Pathogenic Variants and MHC Evolution

**DOI:** 10.3390/ijms23052665

**Published:** 2022-02-28

**Authors:** Pierre Pontarotti, Julien Paganini

**Affiliations:** 1Evolutionary Biology Team, MEPHI, Aix Marseille Université, IRD, APHM, IHU MI, 19-21 Boulevard Jean Moulin, 13005 Marseille, France; 2SNC 5039 CNRS, 13005 Marseille, France; 3Xegen, 15 Rue Dominique Piazza, 13420 Gemenos, France

**Keywords:** MHC evolution, pathogenic escape, variations, immunity, polymorphism

## Abstract

We propose a new hypothesis that explains the maintenance and evolution of MHC polymorphism. It is based on two phenomena: the constitution of the repertoire of naive T lymphocytes and the evolution of the pathogen and its impact on the immune memory of T lymphocytes. Concerning the latter, pathogen evolution will have a different impact on reinfection depending on the MHC allomorph. If a mutation occurs in a given region, in the case of MHC allotypes, which do not recognize the peptide in this region, the mutation will have no impact on the memory repertoire. In the case where the MHC allomorph binds to the ancestral peptides and not to the mutated peptide, that individual will have a higher chance of being reinfected. This difference in fitness will lead to a variation of the allele frequency in the next generation. Data from the SARS-CoV-2 pandemic already support a significant part of this hypothesis and following up on these data may enable it to be confirmed. This hypothesis could explain why some individuals after vaccination respond less well than others to variants and leads to predict the probability of reinfection after a first infection depending upon the variant and the HLA allomorph.

## 1. Introduction: Understanding of the Mechanisms That Lead to MHC Polymorphism

It was realized very early on when the MHC/HLA system was discovered that the system is polymorphic and is a recognition system. Initially, the parasitic hypothesis was put forward. Two properties that could explain resistance to pathogens were considered in the early 1970s [1]. The first mechanism involves a “molecular mimicry” between histocompatibility antigens and viral antigens; if an antigen of a virus has the same specificity as an HLA antigen (called at the time the HLA system). Individuals carrying these particular antigens are not able to develop an immune response to the virus. The second possibility is that, given the histocompatibility, antigens may act as, or interact with, receptor sites on the surface of cells, for the binding of viruses. The maintenance of the polymorphism is explained by the mechanisms of heterozygous advantage and rare allele advantage [1]. A major advance in the understanding of HLA biology was made in 1974 by the demonstration of the MHC plus peptide interaction and the recognition of this complex by T cells [2], although neither the structure of the MHC nor that of T receptors were known.

The gradual discovery of the role of MHC in the immune system and better descriptions of the different HLA allomorphs has allowed to recontextualize the hypothesis of a balanced selection (heterozygous advantage, rare allele advantage) due to a pathogen [3]. The nature of the receptor recognizing the MHC plus peptide complex was elucidated in 1984 [4,5] after a saga that began in the late 1970s (see for review [6]). Another important step was the identification of the structure of MHC proteins, which began in the 1960s [7] and ended with the crystallographic resolution of an MHC protein [8,9] and the identification of the peptide-binding site to the MHC. In addition to the identification of the peptide-binding site, it has been shown by a comparison of different MHC sequences that the variability/polymorphism is located at the peptide-binding site [10]. The next important step was to identify the nature of peptides and their diversity binding to the peptide niche [11], finding that there was a preference for binding to certain peptides based on MHC alleles [12]. Finally, the identification by crystallography of the MHC/Peptide/TCR complex has made it possible to improve the understanding of the interactions between this trio [13,14]. This understanding of the interaction was then quantitative, in particular through the creation of the catalog of immunopeptidomes and its variability according to allomorphs [15]. This has led to support for the reflection on the maintenance and evolution of polymorphism. Current thinking is based on the ability of certain alleles to better recognize certain pathogens [16,17]. It should be noted that the hypotheses explaining the evolution and maintenance of polymorphism via the pathogen driven selection are still based on the same mechanism proposed in the 1970s—the heterozygous advantage, the advantage of rare alleles [1], and in the early 1990s—fluctuating selection [18]. Along with a better understanding of the MHC TCR peptide interaction and the specificity of interaction between HLA allomorphs and peptides, recent years have seen significant advances allowing the establishment of the T cell repertoire by thymic selection (see the seminal article by Kapler et al. [19], and for a relatively recent review, Inglesfiled et al. [20]). This selection is in part dependent on MHC allomorphs. Surprisingly, reflection on the impact of thymic selection is not taken into account in order to understand the role of pathogens on the protective capacity of a given HLA allele and, therefore, on the maintenance and evolution of the MHC. A second important point, which is notably discussed, is that thymic selection does not depend solely on the MHC, but also on the self-peptidome [21]. The third point is the fact that the loci encoding T receptors are variable between individuals [22]. A fourth point is that the development of T receptors is also dependent on stochastic events [23]. A final important point that is not taken into account is the reinfection process and the impact of variants against HLA alleles in this process. We will explain how these processes participate in the mechanisms of maintenance and evolution of polymorphism. In addition to the pathogen-driven evolution of MHC polymorphism/diversity, other mechanisms have been proposed to explain the evolution and maintenance of MHC polymorphism. One of these mechanisms is the possibility that MHC polymorphism/diversity is driven by NK cell receptors. As noted above, TCRs interact with peptides bound by MHC molecules as well as with parts of the peptide-binding domain. Some MHC class I molecules can be ligands for NK and myeloid cells, resulting in another level of selection beyond T cells (it should be noted that this explains, in part, the MHC class I polymorphism but not the MHC class II polymorphism). Indeed, the NK killer inhibitory receptors (KIR) in humans also recognize some of the peptides presented by the MHC molecules, potentially playing a role in the selection of MHC [24]. However, while T cell receptors recognized peptide/MHC in a relatively specific manner; KIRs recognize peptide/MHC complex families in a pattern-based manner [25]. In addition, the number of peptide/MHCs recognized is much lower than in the case of TCR (see, for example, Naiyer et al., 2017 [26]). Thus, although the participation of the NK receptor in the maintenance of the MHC polymorphism may be low, this participation should not be ignored [27]. Another possible hypothesis is the mating choice for dissimilar MHC-driven hypotheses. Disassortative mating could also maintain MHC diversity/polymorphism, a phenomenon that has been described using theoretical analysis [28]. Furthermore, it has been noted that many jawed vertebrates can discern MHC alleles based on olfactory and other cues, and prefer scents of mates with complementary or dissimilar MHC genotypes [29]. These last two agents of selection may participate in the maintenance of the MHC polymorphism and may be complementary to the mechanism of selection directed by the pathogen. In this article, we propose a new perspective on the pathogen driven hypothesis. This revised hypothesis is based on the constitution of the TCR repertoire and on the role of T cell memory and pathogen evolution on the selection and maintenance of MHC polymorphism. It is based on functional observations and gives a rational explanation for the evolution and maintenance of the MHC polymorphism. This hypothesis also better explains the relationship between HLA autoimmune diseases, malignancies and infections (resistance versus sensitivity) [30]. At the end of the article, we provide a test of this hypothesis using the extensive data obtained from mass vaccination against SARS-CoV-2. In this case, vaccination is used as a proxy for viral infection. Therefore this hypothesis can be falsified, which is rare in the field of evolutionary biology [31]. In addition, validation of this hypothesis could make it possible to perform vaccination in a personalized way, and at least predict which people will relapse according to the genomic sequences of the new variants of a given pathogen according to HLA typing.

## 2. Revisiting the Pathogen-Based Hypothesis to Explain MHC Polymorphism/Diversity

### 2.1. Establishing the Naive TCR Repertoire: Genetic and Stochastic Parameters

The role played by MHC on the resistance/susceptibility to pathogens is often discussed from the perspective of their capacity to bind pathogen-derived peptides through the peptide-binding domain and present them to T cells [16]. The peptide plus MHC complex recognized by the TCR is known as the T cell epitope. The aim of this section is to discuss the fact that T cell recognition is not only based on the recognition of the peptides by a given MHC allomorph but also on the availability of the T cell epitope in the TCR repertoire. The T cell repertoire on an individual level is built via proximate causation and ultimate causation [32]. “Proximate causation” is the immediate influence on an outcome: the thymic selection of T cell receptors that can recognize MHC molecules. “Ultimate causation” refers to evolutionary influences on an outcome, for example, the evolution of germline-encoded T cell receptor recognition of MHC molecules. Therefore, at a population level in humans, TCR germinal diversity may be selected (ultimate causation). However, TCR germinal locus diversity is also important for the thymic selection process (see below). In this paper, we will focus only on the proximate causation of the T cell individual repertoire. We will show that thymic selection depends (a) on genetic parameters, and (b) on stochastic parameters.

### 2.2. Genetic Parameters

#### 2.2.1. MHC Allomorph Polymorphism/Self-Peptidome Variability and their Role on Thymic Selection

##### Thymic Selection and MHC Allomorph Polymorphism

MHCs, which are composed of MHC class I and MHC class II genes, are highly polymorphic loci found in all jawed vertebrates. In humans, over 15,000 allotypes are encoded by the five most polymorphic genes [33]. Polymorphism is concentrated around the peptide-binding cleft of the HLA/MHC proteins and, thus, distinct HLA/MHC allotypes have distinct peptide-binding preferences, which are determined by anchor residues that reside within the peptide-binding domain. HLA/MHC variants can each recognize and present to the TCR thousands of peptides, with an overlap between these peptides that is positively correlated to peptide-binding cleft similarity [34]. TCR diversity is generated during T cell ontogeny in the thymus via random rearrangements of variable (V), diversity (D), and joining (J) gene segments from *TCR β* loci, and the random rearrangement of V and J gene segments from the *TCR α* gene loci. This is followed by the selection of αβTCR cells. This process involves immature thymocytes being subjected to both positive and negative selection. The first step is the positive selection resulting in the survival of thymocytes that undergo intra-thymic migration. This step ensures that the thymus produces T cells capable of antigen recognition. This selection takes place in the thymus cortex, where the self-peptide involved in positive selection appears to be generated by the thymoproteasome, a specialized type of proteasome. The second step is the negative selection, which ensures that T cell development produces functional thymocytes that are tolerant to self-antigens. This step occurs in the thymus medulla. Selection is carried out by eliminating high-binding T cells and keeping only T cells that can recognize the HLA/MHC-peptide complex with low-to-medium affinity. Because all genes are likely to be expressed in the thymus tissues, the whole peptidome thus plays a role during both the positive and negative selection steps [20]. The outcome of thymic selection will be different depending upon the allomorphs which present peptides and which give rise to a different naive type in the TCR repertoire. Even if an HLA allomorph is able to recognize a pathogenic peptide, the HLA allomorph plus peptides might not be recognized by any of the TCRs available in the repertoire. Therefore, the HLA/MHC binding properties are not sufficient to confer resistance to a given pathogen, which depends, in addition, on the availability of the TCR epitope after thymic selection. Besides the capacity of the different MHC allomorphs to recognize different peptides, the immunopeptidome differs between individuals, which could lead to differences in the TCR repertoire, despite identical HLA/MHC allomorphs [21]. This idea will be developed in the following section.

#### 2.2.2. Thymic Selection and Self-Immunopeptidome Variability

Variability in the self-immunopeptidome and its impact on thymic selection could explain differences in the TCR repertoire, despite an identical HLA type. The consequence of this difference is shown by the existence of minor histocompatibility antigens (MHAs). MHAs, for example, result in immunological responses during bone marrow transplants between HLA-identical donors and recipients. MHAs correspond to peptides that are polymorphic between these donors and recipients, leading to different epitopes. The outcome of the selection gives rise to a different repertoire. The difference in the TCR repertoire driven by the self-immunopeptidome could be important for the immune response to pathogens [21]. The variability of the self-immunopeptidome (and therefore its impact on the TCR repertoire) could be huge. This variability can be due to single nucleotide polymorphisms (SNPs) differences, which have been well-described for minor antigens [35], but which could correspond to non-homologous regions due to structural variations in individuals’ genomes [36]. In addition, the peptides recognized in particular by class I allomorphs may also come from non-canonical proteins and not only from the annotated proteins available, for example, in UniProt [37,38]. The immunopeptidome includes highly variable sequences such as transposable elements, LINE-1, and endogenous retroviruses [39]. Peng et al. [40] discussed the value of studying diversity in an immunogenomic study of the polymorphic gene involved in the immune system (HLAs, TCRs, KIRS, etc.); the self-immunopeptidome has not been discussed. We propose that self-peptidome diversity must be included in the immunogenomic diversity analysis.

#### 2.2.3. Thymic Selection and Germline TCR Loci Diversity

The germline variability of TCR loci has been demonstrated in several studies. Due to the difficulty of analyzing these complex regions (several paralogues, many of which repeat), the assembly of different haplotypes, and the analysis of variability is a difficult task. As a result, very few analyses have been reported to date concerning the TCRB locus, and they show variability at SNP level as well as structural variability [22,41,42,43]. To the best of our knowledge, structural rearrangements have not been studied for TCRA loci, which are more complex than TCRB, but the variability of SNPs has been described (Scaviner and Lefranc [44] and IMGT database). Furthermore, it is likely that variation between individuals in the number of genes also exists for the TCRA locus. The variability at the germinal level of the TCR locus has been shown to have an impact on the T cell response [45], therefore, the germline variability of the TCR plays a role in the personalized immune response.

### 2.3. Stochastic Determinants and the TCR Repertoire

Many reports show that the TCR naive repertoire of twins is different [23,46]. Evaluation of the preselection diversity of the human T-cell repertoire can be performed via the quantification of VDJ recombinants with out-of-frame DNA sequences, as these sequences are not subject to selection and therefore reflect only the recombination process [47]. This approach has been taken to compare twin TCR repertoires before the thymus selection and has shown that these repertoires are different (see, for example, Heikkilä et al. [46]). This shows that stochasticity plays a role in the establishment of the TCR repertoire and that two individuals with the same genetic background will have a different TCR repertoire.

### 2.4. Correlation with Data: Individuals with the Same MHC Recognize Different Antigenic Peptides

The above sections show that there are many reasons why individuals with the same MHC recognize different antigenic peptides and explain the data obtained in the case of the most studied pathogen, SARS-CoV-2 (Tarke et al. [48], Appendix A). These authors describe a different T repertoire between individuals with the same HLA allomorphs. In fact, different peptide/HLA complexes are recognized by memory T cells. What has not yet been clarified is whether this difference is due to the naive cell pool, as explained above (differences in the immuno-peptidomes in the germinal TCR loci of individuals and their stochastic effects) or whether this is due to different maturation switching from naive T cells to memory T cells [49]. In conclusion, the MHC is important for the recognition of antigenic peptides for a given pathogen, however, even if the MHC allomorph has the capacity of recognizing given peptides, antigenicity is not certain. However, even if MHC only plays a partial role in pathogen protection, we will show that the role of MHC is mandatory, and that the evolution and maintenance of MHC polymorphism/diversity can still be explained by the pathogen driven hypothesis.

## 3. Hypotheses

### 3.1. Naive T Cell Repertoire HLA Dependency Could Explain MHC Polymorphism

HLA resistance to a given pathogen can be explained by the fact that some MHC/HLA allomorphs, even after thymic selection, will give rise to more epitopes that will be recognized by the TCR, since some allomorphs are able to present more peptides from the pathogen [50,51]. It is expected that, in this case, individuals whose TCRs recognize more pathogenic peptide/HLA epitopes will be advantaged and, therefore, the frequency of the best HLA binders will increase in the population after the epidemic due to the pathogen. When another virus arises, other allomorphs could increase or decrease in the population. The penetrance of the resistance with respect to an HLA allomorph in the context of the T cell naive repertoire is not complete (it is only a strong association) [52]. This can be explained in light of thymic selection and the fact that the HLA allomorph plays a partial role in the establishment of the naive TCR repertoire (see above).

This hypothesis is similar to the classical model of pathogen-driven MHC selection (as discussed by Radwan et al., 2020 [16]), except that a layer of complexity is added with the thymus selection.

### 3.2. Pathogen Variant Escaping Memory T Cells as a Motor of MHC Evolution

In this section, we add another layer of complexity by taking into account the evolution of the pathogen and its impact on T cell memory, depending on the HLA allomorph. We have shown above that the resistance capacity of a given HLA allomorph with respect to a given pathogen is explained by the naive T repertoires resulting from thymic selection. We propose here a complementary hypothesis based on the memory of T cells and on the evolution of the pathogen over a generation of host species. A given pathogen, after infection, giving rise to a T cell memory response. This will take place by the maturation of the pool of naive T cells capable of recognizing the HLA and peptides from the pathogen complex. If variants of the pathogen emerge (1), some HLA allomorphs will not be able to recognize the peptides corresponding to the regions where there has been a mutation (or will bind sub-optimally to them), and this will lead to a loss of the corresponding TCR epitopes. The loss of the TCR epitope with regards to variant and specific HLA allomorphs has been described, for example, in the case of SARS-CoV-2 [53,54,55] (Reynolds et al., in Appendix A). Individuals who have lost TCR epitopes in memory T cells will have a greater chance of being reinfected. Indeed, the memory response is established from a pool of naive cells recognizing this peptide. If there are no naive T cells recognizing peptides (or few) other than the mutated one, there will be no immune response (or less) from the T cells and, therefore, no immune protection (or less protection). However, in the case of another individual with the same HLA allomorphs, it is possible that their naive T cells are selected, which recognize unmutated peptides in the variant pathogens. These naive T cells will then be selected as memory cells, and in this case, there will be immune protection. Therefore, with the same HLA allomorph, T immune protection or less immune protection will be possible. It can be concluded that, in any case, allomorphs that recognize the ancestral peptides, rather than the mutated peptide, will be at a disadvantage compared to other allomorphs which do not recognize the ancestral version of the mutated peptide (or which are not impacted by the mutation). Therefore, the frequency of allomorphs that recognize ancestral peptides and not mutated peptides will drop. This will lead to a variation in the allele frequency in the next generation. The allele frequencies could change in the next generation with a new pathogen and similar mechanism, explaining the evolution and maintenance of the MHC polymorphism and diversity (See Appendix A for a simplified model). It should be noted here that the impact of the variant escaping the T cell memory will be stronger for HLA allomorphs that have a lower binding capacity to a given pathogen than those that have higher binding capacity.

### 3.3. Naive Repertoire, Memory Repertoire Pathogen, and HLA Allomorph Frequencies

The preceding paragraphs can be summarized as follows: the frequency of different alleles may vary according to primary infection and the capacity to have a good immune response. It is believed that the allomorphs that recognize a larger number of peptides with adequate affinity [50] will make it possible to have more T cell epitopes and, therefore, better immune protection. As described above, not all HLA epitopes will be recognized by the T cell naive repertoire (due to thymic selection) and, further, by the T cell memory repertoire. However, it seems likely that the best binder to a given allomorph will give advantages to the individuals with them. In the case of reinfection due to the same pathogen variant, the best allomorph binder should still give an advantage. However, since most pathogens mutate and give rise to several variant lineages, two things must be considered. Firstly, the allomorphs that lose the ability to recognize the mutated peptide will have less chance of having memory T cells that recognize the variant. Secondly, the chances will decrease with the number of epitopes recognized by the T cell memory after the first infection. Therefore, the allomorph with the best binding capacity could be the most protective allele, although the other motor that drives MHC selection is virus mutation and its impact on the T cell immunological memory. There is a synergy between the two parts of the hypothesis: weak HLA binders will be less likely to have an adequate adaptive immune response than high HLA binders, and weak HLA binders might be even less protective if the mutations in the variant occurred in antigenic peptides of weak HLA binders. The following classification can be proposed: (1) high HLA binders, the binding of which is not impacted by mutations in the new variants; (2) high HLA binders, the binding of which is impacted by mutations in the new variants; (3) low HLA binders, the binding of which is not impacted by mutations; and (4) low HLA binders, the binding of which is impacted by mutations. The allele hierarchy is clear if we compare Case 1 to Case 4. In Cases 2 or 3, it is more difficult to predict what would be the more protective allele.

## 4. Testing the Hypotheses

This section proposes how to test both potential mechanisms of selection on the evolution and maintenance of MHC polymorphism/diversity. In the case of the hypothesis of pathogenic variants escaping memory T cells as a driver of MHC evolution, we first have to show that the variant reinfection level will be different for the different allomorphs. For the naive T cell repertoire HLA-dependent hypothesis/mechanism, it has to be shown that some alleles protect better than others in the case of a primo infection. In both cases, the frequency of the “bad” allele is lower in the next generation.

### 4.1. Testing the Hypothesis: The Escape of Pathogenic Variants to Memory T Cells as a Driver of MHC Evolution

The aim of the test will be to show that some MHC allomorphs are more impacted by the mutation of a given pathogen than others, i.e., they will lose the binding interaction with the variant peptides. This will, therefore, have an impact on the health of individuals who have these allomorphs and, therefore, on the individual’s fitness (individual reproductive success), leading to a lower frequency of bad allomorphs in the next generation. The test of the hypothesis has to be conducted at two levels. First, it has to be shown that there is a different impact of reinfection for the different allomorphs. The alleles that are more impacted are referred to as “bad” alleles. Secondly, it has to be demonstrated that the frequency of the “bad” alleles is lower in the next generation (this will be developed further in Section 4.3. The impact on the reinfection depending on the HLA allomorph can be tested using SARS-CoV-2 as a paradigm. An approximation of re-infection could also be infection after vaccination. The advantage of using vaccination is that this is performed on a large scale, especially in the case of SARS-CoV-2. At the time of writing this article (21 September 2021), six billion doses of COVID-19 vaccines had been administered across the world [56]. Statistical analysis can, therefore, be carried out on the basis of this data. This will require the constitution of a cohort and its follow-up over time and the identification of certain genetic characteristics that may explain differences in the development of individuals in the cohort. For this, it will be necessary to constitute and analyze a cohort of people infected with the original SARS-CoV-2 strain (Wuhan, China), or people who have been vaccinated with the spike portion of the Wuhan strain, which corresponds to most of the vaccines that are used [57], and to monitor their re-infection. This implies the classification of individuals into two categories: those who do not relapse and those who relapse. Among those who relapse, it will be necessary to carry out a classification according to the variant of the virus at the origin of the relapse. The relapse may take several forms: positive but asymptomatic, positive and slightly ill, and positive and very ill. It will also involve the HLA typing of individuals and a comparison between different groups. In principle, particular HLA allomorphs should be statistically more present in the “relapse” groups. In addition, the HLA allomorphs could differ depending on the variant of the virus responsible for the relapse. It can then be verified that the impact of the mutation on the peptides in their recognition by the allomorphs was predictable [53,54,55], thanks to the analyses and prediction of the peptide/HLA binding using for example NetMHCpan [58], and/or a test for TCR recognition against the variant peptide plus HLA, as described for example by Tarke et al. [48] and Reynolds et al. [55]. This will show that individuals with certain HLA allomorphs will be disadvantaged and will have less chance of survival compared to individuals with other allomorphs and, therefore, the frequency of the allomorphs should change. Reynold et al. [55] reported that healthy individuals receiving one vaccine dose without prior infection showed reduced immunity against variants B.1.1.7 and B.1.351, resulting in increased, abrogated, or unchanged T cell responses, depending on HLA polymorphisms. However, the authors did not demonstrate that in their article. In fact, they showed that, for some alleles, the epitope was lost due to the mutation in the antigenic region in the variant’s peptide by ex vivo analysis and using a mouse model. Therefore, the reduced immune response depending on the HLA allomorph still remains to be tested. If the response to the variant after vaccination depends on HLA polymorphisms, this will be a big step towards demonstrating the escape of pathogenic variants to memory T cells as a driver of the MHC evolution hypothesis. Regarding the memory and pathogen evolution hypothesis it is important to understand the evolutionary mechanism that drives the evolution of a pathogen. This process is well understood in the case of viruses. The evolution of a virus depends firstly on the cellular system used by the virus for its reproduction (replication), including cell surface receptors, translational mechanisms, and nuclear or cytoplasmic structural elements and, secondly, on its capacity to escape the innate and adaptive immune response [59]. Therefore, in Case 1, when a mutation in the virus results in a modification of, for example, the receptor-binding protein of the virus (which allows the virus to enter the interior of the cell), allowing better binding, the fitness of the virus will improve. This improved fitness will give this variant an advantage over the wild-type virus (or ancestral strain) and, therefore, the new variant can replace the old one. The variation could correspond to antigenic peptides recognized by the memory T cell of a given HLA allomorph. The variation gives rise to a loss of binding and, therefore, to a loss of the corresponding epitope of memory T cells. In that case, these allomorphs are referred to as “unlucky” alleles, as this loss of binding is linked to a better adaptation to the host cellular system rather than an immune escape. Besides adaptation to the cellular system, the loss of T cell epitope could be due to immune escape, including adaptive immune escape. This has been shown in the case of chronic viral infection with LCMV, HCV, and HIV for CD8 T cells [60] and for CD4 T cells [61]. Chronic viral infection is a dynamic, metastable equilibrium process, whereas an acute viral infection is an out-of-equilibrium process. During an acute infection, both the host and virus change continuously until either the infection is resolved, it kills the host, or becomes chronic, as in the case of chronic viral infection. Immune evasion can occur in acute infections, in which case a new variant could infect new individuals (as has been seen for SARS-CoV-2 virus infection). If this was the case in acute infections, the most frequent allomorph in a given population would be the one that could be the most affected by the immunological escape.

### 4.2. Testing the Hypothesis: An HLA-Dependent Naive T Cell Repertoire Could Explain MHC Polymorphism

The same kind of test as described in the case of the naive repertoire is also required in this case. This can be carried out by looking at HLA allomorph polymorphism among individuals who respond or do not respond to the vaccine and to investigate whether this is associated with HLA polymorphisms [62,63]. The next step towards consolidating the hypotheses is to test that the frequency of “bad” alleles does indeed decrease in the next generation. This will validate, in part, our hypothesis.

### 4.3. Experimental Design to Test the Change in the Allelic Frequency of an HLA Allomorph after an Outbreak

In the preceding paragraphs, we show that it can be tested whether some HLA allomorphs are more protective than others with regards to a given pathogen. It is likely that the difference in protection by the different MHC allomorphs will be reflected in the difference in frequencies of these alleles, however, this does not constitute proof. Therefore, if we want to go a step further towards consolidating our hypothesis, we have to show a change in allele frequency after an outbreak. As we said in the introduction, the closest thing to such a study is one that was performed in experimental conditions (not natural). The study showed that MHC-II allele frequencies vary in fish populations after exposure to a pathogen [64]. Analysis of the populations before or after major pandemic events should help [65]. For example, the Black Death (1347–1351) killed as much as 30% of the European population, which should have had an impact on the difference in allele frequency before and after the epidemic. Such research is in its infancy. For example, Immel et al. [66] have investigated an MHC and a large panel of other immunity-related genes from 36 individuals discovered in three 16th century mass graves of plague victims in Ellwangen, southern Germany. They compare them to 50 present-day inhabitants of Ellwangen. Even if this may be an approximation of what needs to be tested, it shows that, in the future, such data could be available. To test the escape of pathogenic variants to memory T cells as a driver of the MHC evolution hypothesis, we need information about the possibility that the pathogen has mutated within a species (a human generation). The analysis by Bramanti et al. [67] shows that this is likely to be the case during various plague pandemics. To go further, the in silico analyses of the mutation on the binding capacities of the different MHC allomorphs has to be investigated, as well as the host immune system, as described by Immel et al. [66], in the hope that samples will be available in the future.

## 5. Concluding Remarks

We present here two complementary hypotheses explaining the evolution and maintenance of the MHC polymorphism. This is the first time that these hypotheses have been able to be tested on a large scale, using vaccination against the pathogen responsible for the current SARS-CoV-2 epidemic as a proxy for infection. The COVID-19 pandemic is having a dramatic impact on our societies. But ironically, it also allows spectacular advances, particularly in virology and immunology. It is also the first pandemic for which we have sufficient scientific means to analyze its impact on genetic variability.

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
