# Peer review of "COVID-19 Pandemic: Escape of Pathogenic Variants and MHC Evolution"

_ijms, 2022, doi:10.3390/ijms23052665_

Round 1
Reviewer 1 Report
The paper by Pontarotti et al put forward a hypothesis to explain how pathogenic viral variants may escape immune detection based on the MHC allomorphs they express.
The paper was very long and did not provide a high level of novelty int he message that they were putting forward. The Introduction and sections 2 and 3 were long and the authors should try to reduce these sections. They had to set the scene i understanding prior knowledge around MHC alleles, TCR recognition of MHC alleles but in the end the main message came in section 4.2. The message from that is that we need to do more tests to determine the "bad alleles". But how would do they go about doing that? Section 4.3 attempts to go down the path but gets side tracked on an unrelated disease the plague.
I don’t dispute the evidence that they present but the message was lost in all the background details and so the impact to the reader was lost. Reducing the size of the paper and to articulate how they would test the hypothesis more clearly would go some way toward improving the readability of the article.
There were a number of typos throughout the manuscript- the first one came in the first sentence of the introduction- MCH/HLA.
Author Response
First of all we would like to thank the referees for their comments and insight.
The paper by Pontarotti et al put forward a hypothesis to explain how pathogenic viral variants may escape immune detection based on the MHC allomorphs they express.
The paper was very long and did not provide a high level of novelty in the message that they were putting forward. The Introduction and sections 2 and 3 were long and the authors should try to reduce these sections. They had to set the scene i understanding prior knowledge around MHC alleles, TCR recognition of MHC alleles but in the end the main message came in section 4.2.
We have discussed it a lot while writing the article, but we think the background is very important to understand the hypothesis, please also note that the background presents new idea like for example the self -peptidome as a major player in the establishment of the naive T-cell repertoire
The message from that is that we need to do more tests to determine the "bad alleles". But how would do they go about doing that?
We added a sentence to clarify this issue
Section 4.3 attempts to go down the path but gets side tracked on an unrelated disease the plague.
Our hypothesis is not limited to SARS-CoV-2 but applies to any pathogen.We therefore propose that studies on paleo-epidemics would make it possible to test the hypothesis
I don't dispute the evidence that they present but the message was lost in all the background details and so the impact to the reader was lost. Reducing the size of the paper and to articulate how they would test the hypothesis more clearly would go some way toward improving the readability of the article.
There were a number of typos throughout the manuscript- the first one came in the first sentence of the introduction- MCH/HLA.
The manuscript has been proofread by a professional service to avoid typos and grammatical errors. They are corrected in the new version of the manuscript
Reviewer 2 Report
The authors propose a new hypothesis that explains the maintenance and evolution of the MHC polymorphism and is based on the following features: the constitution of the repertoire of naive T lymphocytes and the evolution of the pathogen and its impact on the immune memory of T lymphocytes. They discuss the impact on the reinfection depending on the MHC allomorph.
Of interest, data from the SARS- CoV-2 pandemic might support a significant part of this hypothesis. The authors speculate that this hypothesis could explain why some individuals after vaccination respond less well than others to variants and leads to predict the probability of reinfection after a first infection depending upon the variant and the HLA allomorph.
The manuscript and author's hypothesis are of interest and of current clinical interest.
The authors should further consider and discuss another important aspect related to SARS- CoV-2 disease and to the relationship between HLA and autoimmunity. It has been recently reported that a significant number of patients with SARS- CoV-2 lung disease develop autoantibodies, in particular, antinuclear antibodies as patients with systemic sclerosis that among its clinical manifestations include pulmonary involvement in the form of a restrictive syndrome secondary to interstitial pneumopathy resembling COVID‐19 interstitial pneumonia. The authors should discuss these important literature data as recently reported (COVID-19 and Immunological Dysregulation: Can Autoantibodies be Useful? Clin Transl Sci. 2021 Mar;14(2):502-508; Antinuclear antibodies in COVID 19. Clin Transl Sci. 2021;14(5):1627-1628. doi:10.1111/cts.13026).
Author Response
First of all we would like to thank the referees for their comments and insight.
The authors propose a new hypothesis that explains the maintenance and evolution of the MHC polymorphism and is based on the following features: the constitution of the repertoire of naive T lymphocytes and the evolution of the pathogen and its impact on the immune memory of T lymphocytes. They discuss the impact on the reinfection depending on the MHC allomorph.
Of interest, data from the SARS- CoV-2 pandemic might support a significant part of this hypothesis. The authors speculate that this hypothesis could explain why some individuals after vaccination respond less well than others to variants and leads to predict the probability of reinfection after a first infection depending upon the variant and the HLA allomorph.
The manuscript and author's hypothesis are of interest and of current clinical interest.
The authors should further consider and discuss another important aspect related to SARS- CoV-2 disease and to the relationship between HLA and autoimmunity. It has been recently reported that a significant number of patients with SARS- CoV-2 lung disease develop autoantibodies, in particular, antinuclear antibodies as patients with systemic sclerosis that among its clinical manifestations include pulmonary involvement in the form of a restrictive syndrome secondary to interstitial pneumopathy resembling COVID‐19 interstitial pneumonia. The authors should discuss these important literature data as recently reported (COVID-19 and Immunological Dysregulation: Can Autoantibodies be Useful? Clin Transl Sci. 2021 Mar;14(2):502-508; Antinuclear antibodies in COVID 19. Clin Transl Sci. 2021;14(5):1627-1628. doi:10.1111/cts.13026).
We have read the article and many articles on the subject, however we do not believe that the "lost memory hypothesis" can explain autoimmune diseases. The relationship between autoimmune diseases and HLA genes could be explained via the naive T cell repertoire, but that is beyond the scope of the manuscript.
Round 2
Reviewer 1 Report
I am happy that the authors have addressed issues I riased in a satisfactory manner